# Construction of Z-Scheme g-C_3_N_4_/CNT/Bi_2_Fe_4_O_9_ Composites with Improved Simulated-Sunlight Photocatalytic Activity for the Dye Degradation

**DOI:** 10.3390/mi9120613

**Published:** 2018-11-22

**Authors:** Lijing Di, Hua Yang, Tao Xian, Xiujuan Chen

**Affiliations:** 1State Key Laboratory of Advanced Processing and Recycling of Non-ferrous Metals, Lanzhou University of Technology, Lanzhou 730050, China; dlj0308@sina.com (L.D.); chenxj@lut.cn (X.C.); 2College of Physics and Electronic Information Engineering, Qinghai Normal University, Xining 810008, China; xiantao1985@126.com

**Keywords:** Bi_2_Fe_4_O_9_, g-C_3_N_4_, CNT, photocatalysis, Z-scheme heterojunction

## Abstract

In this work, ternary all-solid-state Z-scheme g-C_3_N_4_/carbon nanotubes/Bi_2_Fe_4_O_9_ (g-C_3_N_4_/CNT/BFO) composites with enhanced photocatalytic activity were prepared by a hydrothermal method. The morphology observation shows that ternary heterojunctions are formed in the g-C_3_N_4_/CNT/BFO composites. The photocatalytic activity of the samples for the degradation of acid orange 7 was investigated under simulated sunlight irradiation. It was found that the ternary composites exhibit remarkable enhanced photocatalytic activity when compared with bare BFO and g-C_3_N_4_/BFO composites. The effect of the CNT content on the photocatalytic performance of the ternary composites was investigated. The photocatalytic mechanism of g-C_3_N_4_/CNT/BFO was proposed according to the photoelectrochemical measurement, photoluminescence, active species trapping experiment and energy-band potential analysis. The results reveal that the introduction of CNT as an excellent solid electron mediator into the ternary composites can effectively accelerate the electron migration between BFO and g-C_3_N_4_. This charge transfer process results in highly-efficient separation of photogenerated charges, thus leading to greatly enhanced photocatalytic activity of g-C_3_N_4_/CNT/BFO composites. Furthermore, the g-C_3_N_4_/CNT/BFO composites also exhibit highly-efficient photo-Fenton-like catalysis property.

## 1. Introduction

In the past decade, semiconductor photocatalysis has become one of the most promising technologies for environmental purification and solar energy conversion [1,2,3,4,5]. Exploration of visible-light-driven photocatalysts is the research hotspot in photocatalytic field. Bi_2_Fe_4_O_9_ (BFO), as a narrow bandgap semiconductor (~2.1 eV), is demonstrated to be an important photocatalyst for the degradation of organic dyes, phenol, and aqueous ammonia under visible light irradiation [6,7,8,9,10,11,12,13,14]. Besides its photocatalytic behavior, BFO also exhibits pronounced multiferroic property [15,16]. However, the high recombination rate of photoexcited electron–hole (e^−^-h^+^) pairs in bare BFO seriously limits its further application in the photocatalytic field. To overcome this problem, several modification strategies, including decoration with noble metals and creation of type-II heterojunction with other semiconductors, have been developed to promote the separation of photogenerated charges [17,18,19,20,21,22]. Although the recombination of photogenerated charges can be suppressed through the above methods, the redox ability of photogenerated electrons and/or holes could be more or less weakened simultaneously [23,24].

Recently, construction of direct all-solid-state Z-scheme heterojunction has attracted considerable attention because it can not only enhance the separation of photogenerated charges but also maintain their high redox ability [25,26,27,28,29,30]. A typical direct Z-scheme heterostructured photocatalyst consists of two semiconductors with overlapping energy-band potentials. Under light irradiation, the photoinduced electrons from one semiconductor with less-negative conduction band (CB) will recombine with the photoinduced holes from another semiconductor with less-positive valence band (VB) at the heterostructure interfaces. Thus, the photogenerated electrons and holes with stronger reduction and oxidation abilities are separately maintained at different semiconductors. This is beneficial for the enhancement of photocatalytic efficiency [25,26,27,28,29,30]. To date, graphitic carbon nitride (g-C_3_N_4_) based direct Z-scheme photocatalysts have been extensively studied since they are able to achieve excellent photocatalytic performances [31,32,33,34,35,36]. g-C_3_N_4_, as a typical metal-free polymeric semiconductor, is regarded as a promising photocatalyst for water splitting and degradation of pollutants under visible light irradiation [37,38]. It exhibits many remarkable properties such as high thermochemical stability, unique electronic structure, appropriate bandgap of ~2.7 eV, more negative energy-band potentials and facile fabrication via simple heat treatment [37,38]. These advantages make g-C_3_N_4_ an ideal candidate in the construction of efficient direct Z-scheme photocatalysts with other semiconductors. It is found that BFO possesses appropriate energy-band potentials that match with those of g-C_3_N_4_, implying that they are suitable to form promising direct Z-scheme composite photocatalysts. It was reported that g-C_3_N_4_/BFO composites exhibited enhanced photocatalytic activity for the degradation of rhodamine B (RhB) [36]. However, there is no work concerned the interfacial environment of heterojunctions, which generally has an important effect on the charge migration.

Generally, the interfacial environment of direct Z-scheme photocatalysts is complicated, which may result in low migration rate or opposite accumulation of photogenerated charges [39,40]. To achieve an excellent photocatalytic activity of direct Z-scheme photocatalysts, their interfacial property needs to be further improved. The introduction of electron mediators (such as noble metals, quantum dots, graphene and carbon nanotubes) at the interface between the two semiconductors of direct Z-scheme photocatalytic system has been considered as a convenient way to modify the interfacial property [41,42,43,44,45,46]. In this case, these mediators can act as an electron transfer channel, thus promoting the migration of electrons across the interface [41,42,43,44,45,46]. Therefore, a significantly enhanced photocatalytic activity can be achieved. Particularly, carbon nanotubes (CNT) have been frequently used as electron mediators owing to their unique one-dimensional structure and outstanding electron transport property [44,45,46]. To the best of our knowledge, however, no work has been focused on the photocatalytic performance of Z-scheme g-C_3_N_4_/BFO photocatalysts with CNT electron mediators. In this work, we prepared novel Z-scheme g-C_3_N_4_/CNT/BFO composite photocatalysts. Their enhanced photocatalytic activity was evaluated by the degradation of acid orange 7 (AO7) under simulated sunlight irradiation. The migration behavior of photogenerated charges and involved photocatalytic mechanism were investigated and discussed in detail.

## 2. Materials and Methods

### 2.1. Fabrication of g-C_3_N_4_ Nanoparticles

g-C_3_N_4_ nanoparticles were fabricated by direct heat treatment of melamine, followed by H_2_SO_4_ treatment [47]. In a typical preparation process, 5 g of melamine were placed into a corundum boat with a cover, and then heated at 520 °C for 4 h. The obtained yellow product was ground into powder and collected for the next treatment. One gram of the powder was dispersed into 50 mL of H_2_SO_4_ (98 wt%), and magnetically stirred for 10 h and then sonicated for another 10 h. After that, the suspension was placed for 12 h to obtain the flocculent precipitate, and the supernatant was removed. The precipitate was washed with distilled water repeatedly until the pH value of solution became neutral, and then dried at 60 °C for 12 h to obtain final g-C_3_N_4_ nanoparticles.

### 2.2. Preparation of g-C_3_N_4_/BFO Composites

BFO nanoplatelets used in the study were prepared by a hydrothermal method as described in the literature [48]. For the preparation of g-C_3_N_4_/BFO composites, stoichiometric amounts of g-C_3_N_4_ and BFO were dispersed into 40 mL of distilled water under magnetically stirred for 5 h. The mixture was filled into a Teflon-lined autoclave and heat-treated at 130 °C for 3 h. After the completion of the hydrothermal reaction, the product was collected by centrifugation, washed repeatedly with distilled water and dried at 70 °C for 8 h. Based on this method, a series of g-C_3_N_4_/BFO composites with different mass ratios of g-C_3_N_4_ were prepared. The composites prepared at *m*_g-C_3_N_4__/(*m*_g-C_3_N_4__ + *m*_BFO_) × 100% = 2%, 4%, 8% and 16% were designated as 2%g-C_3_N_4_/BFO, 4%g-C_3_N_4_/BFO, 8%g-C_3_N_4_/BFO and 16%g-C_3_N_4_/BFO, respectively.

### 2.3. Preparation of g-C_3_N_4_/CNT/BFO Composites

To prepare ternary g-C_3_N_4_/CNT/BFO composites, a stoichiometric amount of CNT was added into 40 mL of distilled water with ultrasonic treatment for 5 h to form uniform suspension. BFO and g-C_3_N_4_ with a mass ratio of 92:8 were loaded into the CNT suspension under vigorous stirring for 5 h. The following hydrothermal treatment, washing and drying procedure was similar to that for g-C_3_N_4_/BFO preparation. According to the above procedure, a series of g-C_3_N_4_/CNT/BFO composites with different mass contents of CNT in the composites (0.5%, 1%, 1.5% and 3%) was fabricated. The corresponding samples were named as 8%g-C_3_N_4_/0.5%CNF/BFO, 8%g-C_3_N_4_/1%CNT/BFO, 8%g-C_3_N_4_/1.5%CNT/BFO and 8%g-C_3_N_4_/3%CNT/BFO, respectively.

### 2.4. Photocatalytic Experiment

AO7 was employed as the target pollutant to evaluate the photocatalytic degradation activity of the as-prepared samples under simulated sunlight irradiation of a 300 W xenon lamp. In a typical process, the initial AO7 concentration was 5 mg L^−1^ with a photocatalyst loading of 0.5 g L^−1^. Prior to irradiation, the reaction solution was continuously stirred in the dark for 0.5 h to reach the adsorption–desorption equilibrium between the photocatalyst and AO7 molecules. During the photocatalytic reaction, a small amount of the suspension was sampled at a certain time interval and centrifuged to remove the photocatalyst. The absorbance of the solution at 484 nm was recorded to measure the concentration of AO7 by an ultraviolet-visible (UV-vis) spectrophotometer.

To investigate the photocatalytic reusability of the photocatalysts, they were collected by centrifugation after the first photocatalytic experiment. The recycled photocatalysts were washed with distilled water and dried at 70 °C for 10 h in an oven. Then, the recovered photocatalysts were added into the fresh AO7 solution for the next cycle of the photocatalytic experiment.

The active species generated in the photocatalytic reaction were detected by the trapping experiment. To achieve this aim, ammonium oxalate (AO, 2 mmol L^−1^), isopropanol (IPA, 2 mmol L^−1^) and benzoquinone (BQ, 1 mmol L^−1^) were used as the scavengers of photogenerated h^+^, hydroxyl (•OH) and superoxide (•O_2_^−^), respectively [49]. The scavenging experimental process followed the photocatalytic procedure as described above.

### 2.5. Electrochemical Measurement

The electrochemical workstation (CHI 660E, Shanghai Chenhua Instrument Co. Ltd, Shanghai, China) with a three-electrode cell was used to measure the photocurrent response and electrochemical impedance spectroscopy (EIS). In the three-electrode system, a platinum foil and a standard calomel electrode were employed as the counter and reference electrodes, respectively. The working electrode was fabricated according to the following process. First, 15 mg photocatalysts, 0.75 mg carbon black and 0.75 mg polyvinylidene fluoride (PVDF) were introduced into 1-methyl-2-pyrrolidione (NMP) to obtain slurry. Then, the slurry was uniformly coated on a 1.0 × 1.0 cm^2^ fluoride-doped tin oxide glass electrode. Finally, the working electrode was obtained after drying at 60 °C for 5 h in an oven. The measurement system was irradiated by a 300 W Xe lamp. A 0.1 mol L^−1^ Na_2_SO_4_ solution was employed as the electrolyte. The photocurrent–time (I-t) curves were tested at a fixed bias potential of 0.2 V. The EIS test was performed by using a sinusoidal voltage pulse with amplitude of 5 mV and in the frequency range from 10^−2^ to 10^5^ Hz.

### 2.6. Characterization

The phase purity of the samples was examined by X-ray diffraction (XRD) on a D8 Advance x-ray diffractometer (Bruker AXS, Karlsruhe, Germany) and Fourier transform infrared spectroscopy (FTIR) on an IFS 66v/s infrared spectrophotometer (Bruker, Karlsruhe, Germany). The morphology of the samples was observed by field-emission scanning electron microscopy (SEM) on a JSM-6701F scanning electron microscope (JEOL Ltd., Tokyo, Japan) and field-emission transmission electron microscopy (TEM) on a JEM-1200EX transmission electron microscope (JEOL Ltd., Tokyo, Japan). The UV-vis diffuse reflectance spectra (DRS) of the samples were measured using a TU-1901 double beam UV-Vis spectrophotometer (Beijing Purkinje General Instrument Co. Ltd., Beijing, China). The X-ray photoelectron spectroscopy (XPS) was used to measure the chemical composition and chemical state of elements on a PHI-5702 multi-functional x-ray photoelectron spectrometer (Physical Electronics, hanhassen, MN, USA). The Brunauer–Emmett–Teller (BET) surface area and pore size distribution of the samples were measured by N_2_ adsorption–desorption technique using a Tristar II 3020 porosimetry analyzer (Micromeritics Instrument Corporation, Norcross, GA, USA). Fluorescence spectrophotometer (Shimadzu RF-6000, Kyoto, Japan) was employed to record the photoluminescence (PL) spectra of the samples with an excitation wavelength of 410 nm.

## 3. Results and Discussion

### 3.1. XRD and FTIR Analysis

Figure 1 shows the XRD patterns of g-C_3_N_4_, BFO, 16%g-C_3_N_4_/BFO and 8%g-C_3_N_4_/3%CNT/BFO. An obvious diffraction peak at 27.4° is found in the XRD pattern of g-C_3_N_4_ nanoparticles, which is assigned to the (002) crystal plane of graphite-like carbon nitride. Similar result was also observed in the g-C_3_N_4_ nanosheets reported previously [50]. For BFO, all diffraction peaks can be indexed according to the standard pattern of the orthorhombic phase BFO (PDF#74-1098). The XRD patterns of 16%g-C_3_N_4_/BFO and 8%g-C_3_N_4_/3%CNT/BFO are similar to that of bare BFO, indicating that the phase structure of BFO in the composites undergo no change. In addition, the characteristic diffraction peaks of g-C_3_N_4_ and CNT are not found in the XRD patterns of the composites, which are mainly attributed to their weak X-ray diffraction capabilities and low content in the composites.

Figure 2 shows the FTIR spectra of g-C_3_N_4_, BFO, 16%g-C_3_N_4_/BFO and 8%g-C_3_N_4_/3%CNT/BFO. In the spectrum of g-C_3_N_4_, the peak located at ~808 cm^−1^ is attributable to the bending vibration of C‒N heterocycles. A series of absorption bands from 1200 to 1600 cm^−1^ are related to the stretching vibration of C‒N. The band at 1644 cm^−1^ corresponds to the C=N stretching vibration. For BFO, the obvious absorption bands at about 812 and 605 cm^−1^ can be assigned to the Fe‒O stretching and Fe‒O‒Fe bending vibrations in the FeO_4_ tetrahedron, respectively [36]. The absorption band for the O‒Fe‒O bending vibration in the FeO_4_ tetrahedron is detected at ~538 cm^−1^ [36]. The broad absorption bands centered around 480 and 439 cm^−1^ originate from Fe-O stretching vibration in the FeO_6_ octahedron [51]. In the case of the composites, all the characteristic absorption bands of g-C_3_N_4_ and BFO appear in their spectra, confirming the coexistence of g-C_3_N_4_ and BFO in the composites. No traces of typical absorption bands for CNT are detected in the spectrum of 8%g-C_3_N_4_/3%CNT/BFO. This is probably due to the fact that the characteristic absorption bands of CNT are covered by the absorption bands of g-C_3_N_4_ [52].

### 3.2. BET Analysis

Figure 3 presents the N_2_ adsorption–desorption isotherm of 8%g-C_3_N_4_/3%CNT/BFO and its pore size distribution plot (inset). According to the International Union of Pure and Applied Chemistry (IUPAC) classification, the isotherm belongs to the type IV with a hysteresis loop at higher relative pressure, indicating the presence of mesoporous structure. The inset of Figure 3 shows the pore-size distribution curves derived from the adsorption branch of the isotherm using the Barrett–Joyner–Halenda (BJH) method [53], implying the possible existence of micropores of 60–160 nm in the composite. The tail out to extremely large pore could not be caused by physical property. The BET surface area of the composite is obtained to be ~8.1 m^2^/g.

### 3.3. XPS Analysis

The surface chemical compositions and chemical states of BFO and 8%g-C_3_N_4_/1.5%CNT/BFO were investigated by XPS. Figure 4a shows the high-resolution XPS spectra of Bi 4f. In the case of BFO, the strong peaks at 164.0 eV and 158.6 eV belong to the Bi 4f_5/2_ and Bi 4f_7/2_ of Bi^3+^, respectively [54,55]. For the composite, a slight shift of the Bi 4f binding energy peaks is observed, which could be attributed to the interaction of BFO with g-C_3_N_4_ and CNTs. The high-resolution XPS spectra of Fe 2p are presented in Figure 4b. For bare BFO, the broad peak for Fe 2p_3/2_ can be deconvoluted into two peaks at 711.5 and 709.8 eV, which are assigned to Fe^3+^ and Fe^2+^, respectively [56,57]. Another main peak at 723.9 eV corresponds to the Fe 2p_1/2_ binding energy. The peak at 718.2 eV is characterized as the corresponding satellite of Fe 2p_3/2_ [58]. For the composite, the Fe 2p peaks exhibit a slight shift similar to that observed for Bi 4f peaks. Figure 4c shows the O 1s high-resolution XPS spectra. The O 1s binding energy for the lattice oxygen of BFO is observed at 529.7 eV. The peaks at 531.5 (BFO) and 531.8 eV (8%g-C_3_N_4_/1.5%CNT/BFO) are probably attributed to the surface defects and chemisorbed oxygen species [59]. On the high-resolution C 1s spectrum in Figure 4d, the peak at 288.2 eV is caused by the sp^2^-hybridized carbon atom in the g-C_3_N_4_, and the peak at 284.6 eV belongs to the C-C bond in CNT. The N 1s spectrum (Figure 4e) can be divided into three peaks located at ~398.3, ~398.9 and ~401.0 eV. It is worth noting that the characteristic peak of C=N–C bonding in g-C_3_N_4_ is split into two peaks (i.e., ~398.3 and ~398.9 eV), indicating the interaction between g-C_3_N_4_ and CNT. This phenomenon was also found in the CNT/g-C_3_N_4_ hybrid film [60]. The peak at ~401.0 eV is associated with the N atoms in N(-C)_3_ [61].

### 3.4. Morphology Observation

Figure 5a shows the SEM image of bare BFO. It is seen that the BFO sample is composed of platelet-like particles with the edge length of several hundred nanometers and the particles have a smooth surface. As seen from the SEM image of Figure 5b, g-C_3_N_4_ is formed into irregular nanoparticles with 25‒40 nm in diameter. The SEM image of 8%g-C_3_N_4_/BFO in Figure 5c reveals that BFO nanoplates are decorated by g-C_3_N_4_ nanoparticles. From the TEM image of 8%g-C_3_N_4_/1.5%CNT/BFO (Figure 5d), it can be seen that g-C_3_N_4_ nanoparticles and BFO nanoplates are connected with CNT, indicating the formation of ternary g-C_3_N_4_/CNT/BFO composites. Notably, the ternary composites are treated with ultrasound for 10 min before the TEM observation and undergo no destruction, which reveals the tight combination of g-C_3_N_4_, CNT and BFO in the composites.

To further demonstrate the formation of ternary heterojunctions, the elemental mapping was used to observe the 8%g-C_3_N_4_/1.5%CNT/BFO composite, as shown in Figure 6. The dark field scanning TEM (DF-STEM) image of the composite is shown in Figure 6a. Figure 6b–f presents the corresponding elemental maps of the selected region in Figure 6b (marked by yellow outline). One can see that the nanotubes show the elemental distribution of C and are therefore confirmed to be CNT. The small-sized particles integrated with CNT display the elemental distribution of N, confirming that they are g-C_3_N_4_ nanoparticles. The nanoplates anchored onto CNT show the elemental distribution of Bi, Fe and O, revealing that they are BFO nanoplates. The result further confirms the formation of ternary g-C_3_N_4_/CNT/BFO heterostructures.

### 3.5. Optical Absorption Property

Figure 7a shows the UV-vis diffuse reflectance spectra of g-C_3_N_4_, BFO and g-C_3_N_4_/BFO composites. To precisely confirm the absorption edge of the samples, the corresponding first-derivative curves of the UV-vis diffuse reflectance spectra are shown in Figure 7b, where the peak wavelength is determined to be the absorption edge of the samples [62]. It is seen that the absorption edge of g-C_3_N_4_ nanoparticles is located at ~430 nm, and their corresponding bandgap energy (*E*_g_) is calculated to be ~2.88 eV. BFO nanoplates exhibit an absorption edge at about 550 nm, from which their bandgap is estimated to be ~2.25 eV. It is worth noting that no obvious shift for the absorption edges of BFO and g-C_3_N_4_ is detected in the g-C_3_N_4_/BFO composites, which suggests that their bandgap energies do not undergo obvious change.

Figure 8a presents the UV-vis diffuse reflectance spectra of g-C_3_N_4_/CNT/BFO composites, and the corresponding first derivative spectra are shown in Figure 8b. In Figure 8a, one can see that the ternary g-C_3_N_4_/CNT/BFO composites exhibit obvious enhanced absorption in the whole wavelength range compared with g-C_3_N_4_/BFO composites, and moreover the light absorbance of the ternary composites gradually increases with the increase of CNT content. This is mainly ascribed to the strong light absorption of CNT in the UV-visible light region. As shown in Figure 8b, the absorption edges of g-C_3_N_4_ and BFO in the ternary composites are almost unchanged, suggesting that there is no impact of CNT on the bandgap energies of g-C_3_N_4_ and BFO.

### 3.6. Photocatalytic Activity

Figure 9 shows the photocatalytic performance of g-C_3_N_4_, BFO and g-C_3_N_4_/BFO composites toward the degradation of AO7 under simulated sunlight irradiation. Before the photocatalysis, the blank and adsorption experiments were also carried out, revealing that the self-degradation and adsorption percentage of AO7 are relatively small. It is seen that bare BFO nanoplates exhibit low photocatalytic activity due to the high recombination rate of photogenerated electron–hole pairs. After combination of BFO with g-C_3_N_4_, the photocatalytic activity of resulted g-C_3_N_4_/BFO composites increases with increasing the content of g-C_3_N_4_ and achieves the highest value for 8%g-C_3_N_4_/BFO. The improved photocatalytic activity is mainly attributed to the formation of Z-scheme heterojunctions between g-C_3_N_4_ and BFO. However, further increase in the content of g-C_3_N_4_ results in a decreased photocatalytic performance of the composites. This phenomenon is probably due to the fact that the formed g-C_3_N_4_/BFO heterojunctions could be reduced by excessive load of g-C_3_N_4_.

The simulated sunlight photocatalytic behavior of g-C_3_N_4_/CNT/BFO composites was also investigated by the degradation of AO7, as shown in Figure 10. As expected, the photocatalytic activity of the composites is significantly enhanced by the introduction of CNT. The photocatalytic activity of the composites follows the sequence: 8%g-C_3_N_4_/1.5%CNT/BFO > 8%g-C_3_N_4_/3%CNT/BFO > 8%g-C_3_N_4_/1%CNT/BFO > 8%g-C_3_N_4_/0.5%CNT/BFO > 8%g-C_3_N_4_/BFO. It is found that the optimum CNT loading content is 1.5%. Nevertheless, the photocatalytic efficiency of the composites begins to decrease when the content of CNT is higher than 1.5%. The possible reason is that the excessive CNT may shield the light, thus decreasing the photon absorption of the photocatalysts.

It is generally accepted that the recyclability of photocatalysts is an important factor for their photocatalytic applications. Figure 11a shows the recycling photocatalytic degradation of AO7 over 8%g-C_3_N_4_/1.5%CNT/BFO composite under the same conditions. After three cycles, the composite does not show a significant decrease in the photocatalytic activity, suggesting that the as-prepared composite exhibits good photocatalytic reusability. The structure of the composite after successive degradation experiments was examined by XRD. Figure 11b presents the XRD patterns of 8%g-C_3_N_4_/1.5%CNT/BFO before and after the photocatalytic reaction, indicating no apparent structural changes.

### 3.7. Photo-Fenton-Like Catalytic Activity

Besides the photocatalytic activity, the photo-Fenton-like catalysis properties of products were also evaluated. Figure 12 shows the degradation of AO7 over BFO, 8%g-C_3_N_4_/BFO and 8%g-C_3_N_4_/1.5%CNT/BFO under simulated sunlight irradiation and in the presence of H_2_O_2_. On the addition of H_2_O_2_ into the reaction solution, the degradation activities of the samples are much enhanced. Among these samples, 8%g-C_3_N_4_/1.5%CNT/BFO possesses the highest photo-Fenton-like catalysis activity. During the photo-Fenton-like catalysis process, Fe^2+^ on the catalyst surface reacts with H_2_O_2_ to form •OH radicals and simultaneously Fe^2+^ is converted to Fe^3+^. On the other hand, the photogenerated electrons reduce Fe^3+^ into Fe^2+^, and thus the photo-Fenton-like catalysis process can proceed circularly. It is obvious that the photo-Fenton process leads to increased generation of highly reactive •OH, thus promoting the dye degradation. In the case of 8%g-C_3_N_4_/1.5%CNT/BFO, the separation of photogenerated electrons and holes is further promoted, which accelerates the redox cycle of Fe^3+^/Fe^2+^. As a result, the 8%g-C_3_N_4_/1.5%CNT/BFO composite manifests the highest degradation of the dye.

### 3.8. Photogenerated Charge Separation

Photoelectrochemical measurements and PL spectra were used to evaluate the separation and transfer of photoexcited electrons and holes in the samples. Figure 13a shows the photocurrent responses curves of BFO, 8%g-C_3_N_4_/BFO and 8%g-C_3_N_4_/1.5%CNT/BFO under intermittent irradiation of simulated sunlight for several on–off cycles. All samples display obvious photocurrent when the light is switched on, and reproducible photocurrent responses can be observed for the on-off intermittent irradiation. It is seen that the photocurrent of 8%g-C_3_N_4_/BFO is superior to that of bare BFO, revealing that the g-C_3_N_4_/BFO heterojunction suppresses the recombination of photoinduced charges. Remarkably, 8%g-C_3_N_4_/1.5%CNT/BFO exhibits much higher photocurrent than 8%g-C_3_N_4_/BFO, which indicates that the introduction of CNT into the composites leads to more effective separation of photogenerated charges. Figure 13b shows the EIS spectra of BFO, 8%g-C_3_N_4_/BFO and 8%g-C_3_N_4_/1.5%CNT/BFO under simulated sunlight irradiation. It can be seen that all samples exhibit a typical semicircle shape. Generally, the radius of the semicircle is related to charge-transfer resistance at the electrode–electrolyte interface [63]. Among these samples, 8%g-C_3_N_4_/1.5%CNT/BFO exhibits the smallest semicircle radius, suggesting that the ternary composite possesses the lowest charge-transfer resistance and the fastest photogenerated charge migration across the interface. Figure 13c presents the PL spectra of the samples. Obvious PL emission peaks at ~615 nm are detected for all samples. The PL emission peaks are ascribed to the recombination of the photogenerated electrons and holes [64]. Compared with BFO and 8%g-C_3_N_4_/BFO, 8%g-C_3_N_4_/1.5%CNT/BFO has the relatively weak PL intensity, which further reveals the construction of ternary g-C_3_N_4_/CNT/BFO composites is beneficial to the separation of photoexcited charges.

### 3.9. Active Species Analysis

To evaluate the active species and explore the possible reaction mechanism in the 8%g-C_3_N_4_/1.5%CNT/BFO photocatalytic system, AO, BQ and IPA were separately added into the photocatalytic system and used as the scavengers for h^+^, •O_2_^−^ and •OH, respectively [49]. As shown in Figure 14, the introduction of AO slightly suppresses the degradation rate of the dye, revealing that h^+^ exhibits a minor role during the photocatalytic process. In contrast, the photocatalytic activity of composite is remarkably hindered on the addition of IPA, which indicates that •OH plays a crucial role in the photocatalytic reaction. In addition, when BQ is introduced, an apparent decrease in the degradation percentage of AO7 is observed. This result implies that •O_2_^−^ is also an important active species responsible for the dye degradation.

### 3.10. Photocatalytic Mechanism

Generally, the redox ability and migration of photogenerated charges of photocatalysts strongly depend on their energy-band potentials. In this study, the VB and CB of g-C_3_N_4_ and BFO are estimated according to the following equation [65]:*E*_VB_ = *X − E*^e^ + 0.5*E*_g_(1)
*E*_CB_ = *X* − *E*^e^ − 0.5*E*_g_(2)
where *X* is defined as the absolute electronegativity of semiconductors and can be calculated from the arithmetic mean of the electron affinity and the first ionization of the constituent atoms. *E*^e^ is the energy of free electrons on the hydrogen scale (~ 4.5 eV). The *X* value of BFO and g-C_3_N_4_ is calculated to be ~5.92 and ~4.72 eV, respectively. As a result, the CB and VB potentials of BFO are estimated to be +0.29 and +2.54 V vs. normal hydrogen electrode (NHE), respectively. The CB and VB of g-C_3_N_4_ are calculated to be −1.22 and +1.66 V vs. NHE, respectively.

Due to the matching energy-band potentials between g-C_3_N_4_ and BFO, it is expected that g-C_3_N_4_ is incorporated with BFO to form Z-scheme g-C_3_N_4_/BFO heterojunction, as shown in Figure 15. Under simulated sunlight irradiation, both g-C_3_N_4_ and BFO are excited to generate e^−^ and h^+^. The photogenerated e^−^ on the CB of BFO will migrate to the VB of g-C_3_N_4_ across their interface and combine with h^+^ photogenerated from g-C_3_N_4_. Consequently, more photogenerated h^+^ in BFO and e^−^ in g-C_3_N_4_ are able to participate in the photocatalytic reaction. More importantly, the accumulated holes in the VB of BFO and electrons in the CB of g-C_3_N_4_ maintain high oxidation and reduction capabilities, respectively. However, the complicated interfacial environment in the Z-scheme composite photocatalysts could result in the low migration rate or opposite accumulation of photogenerated charges [39,40]. To overcome these disadvantages and further improve the photocatalytic efficiency of Z-scheme g-C_3_N_4_/BFO composites, CNT acting as an efficient electron mediator is introduced at the interface between g-C_3_N_4_ and BFO.

There are two kinds of possible decoration position of CNT in the ternary g-C_3_N_4_/CNT/BFO composites: (i) on the surface of g-C_3_N_4_ or BFO (Figure 16a); and (ii) between g-C_3_N_4_ and BFO (Figure 16b). If CNT is decorated on the surface of g-C_3_N_4_ or BFO, the photogenerated electrons in the CB of g-C_3_N_4_ and BFO will transfer to CNT because the Fermi level of CNT (+0.44 V vs. NHE) is positive to the CB potential of the two semiconductors [46]. However, the electrons on CNT cannot react with O_2_ to form •O_2_^−^ since the Fermi level of CNT is more positive than the redox potential of O_2_/•O_2_^−^ (−0.13 vs. NHE) [66]. Considering •O_2_^−^ is an important active species in the photocatalytic reaction (see Figure 14), this hypothesis is not consistent with the result of the trapping experiment. Therefore, the most reasonable position of CNT is located between C_3_N_4_ and BFO (Figure 16b). In this case, CNT can be served as electron mediator to enhance the migration rate of photogenerated electrons from BFO to g-C_3_N_4_. The fast electron transfer process promotes the recombination of photogenerated electrons in the CB of BFO with photogenerated holes in the VB of g-C_3_N_4_, which further improves the separation rate of photogenerated charges in the g-C_3_N_4_/CNT/BFO composites. This is confirmed by the photoelectrochemical and PL measurements (see Figure 13). On the other hand, compared with the redox potential of O_2_/•O_2_^−^ (−0.13 V vs. NHE) and OH^−^/•OH (+1.89 V vs. NHE) [67], the CB electrons in g-C_3_N_4_ and VB holes in BFO have enough reduction and oxidation capabilities to produce •O_2_^−^ and •OH, respectively. This is consistent with the active species trapping experiment results (see Figure 14).

## 4. Conclusions

In summary, ternary Z-scheme g-C_3_N_4_/CNT/BFO composite photocatalysts have been successfully constructed by a simple hydrothermal route. The photocatalytic experiment demonstrates that the g-C_3_N_4_/CNT/BFO composites exhibit remarkable enhanced photocatalytic activity compared with bare BFO and g-C_3_N_4_/BFO composites. In the ternary Z-scheme photocatalysts, CNT acts as an excellent solid charge mediator between BFO and g-C_3_N_4_, which can promote the separation and migration of photogenerated charges and achieve efficient photocatalytic activity of the ternary nanocomposites. In addition, the good photo-Fenton-like catalysis property of g-C_3_N_4_/CNT/BFO is also found.

## Figures and Tables

**Figure 1 micromachines-09-00613-f001:**
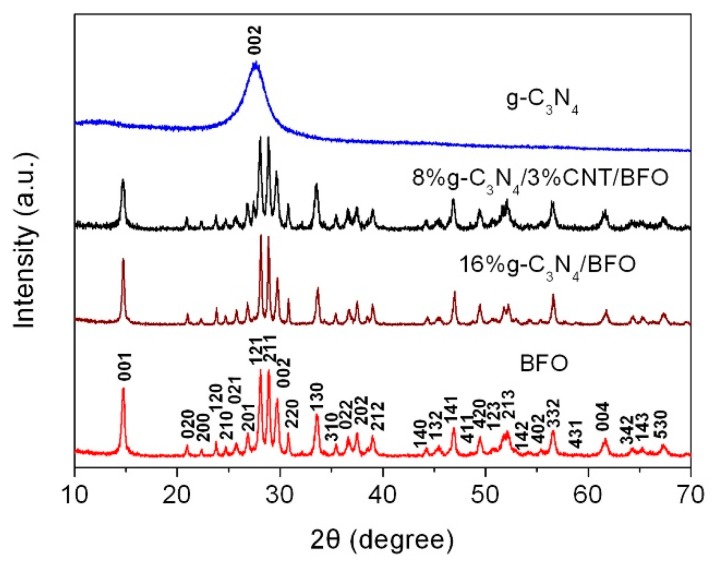
XRD patterns of BFO, g-C_3_N_4_, 16%g-C_3_N_4_/BFO and 8%g-C_3_N_4_/3%CNT/BFO.

**Figure 2 micromachines-09-00613-f002:**
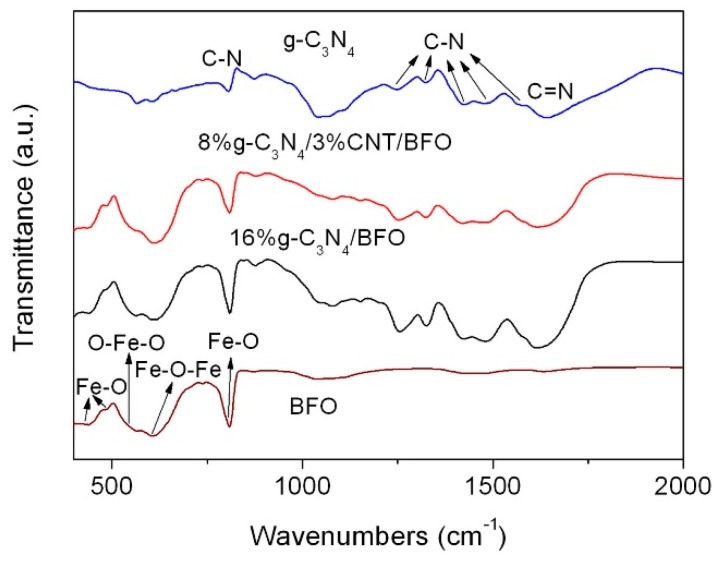
FTIR spectra of BFO, g-C_3_N_4_, 16%g-C_3_N_4_/BFO and 8%g-C_3_N_4_/3%CNT/BFO.

**Figure 3 micromachines-09-00613-f003:**
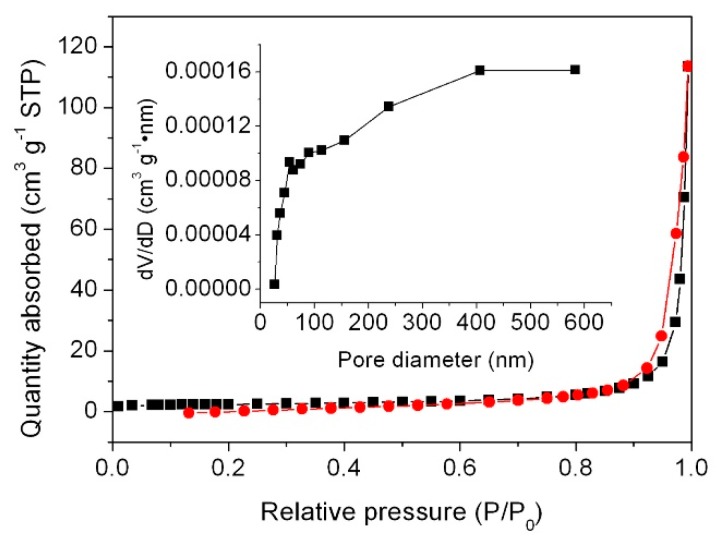
N_2_ adsorption–desorption isotherm and pore size distribution plot of 8%g-C_3_N_4_/1.5%CNT/BFO.

**Figure 4 micromachines-09-00613-f004:**
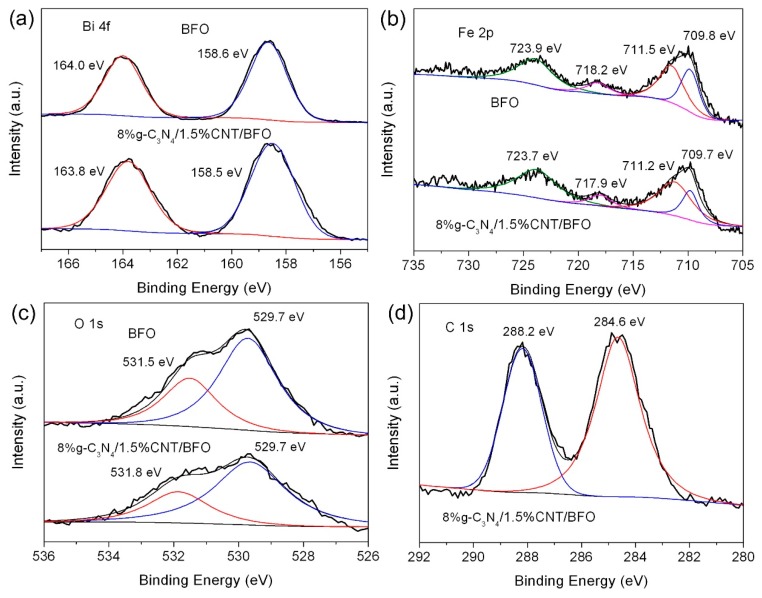
High-resolution XPS spectra of BFO and 8%g-C_3_N_4_/1.5%CNT/BFO: (**a**) Bi 4f XPS spectra; (**b**) Fe 2p XPS spectra; (**c**) O 1s XPS spectra; (**d**) C 1s XPS spectrum; and (**e**) N 1s XPS spectrum.

**Figure 5 micromachines-09-00613-f005:**
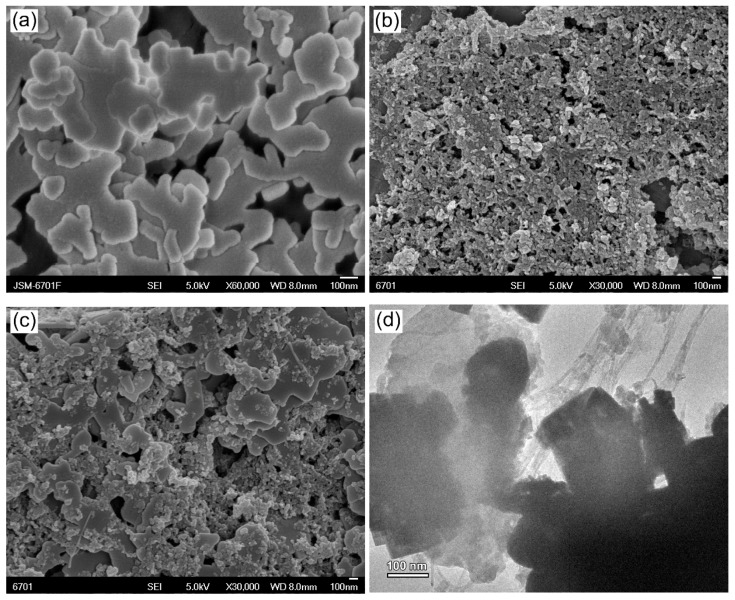
(**a**–**c**) SEM images of BFO, g-C_3_N_4_ and 8%g-C_3_N_4_/BFO, respectively; and (**d**) TEM image of 8%g-C_3_N_4_/1.5%CNT/BFO.

**Figure 6 micromachines-09-00613-f006:**
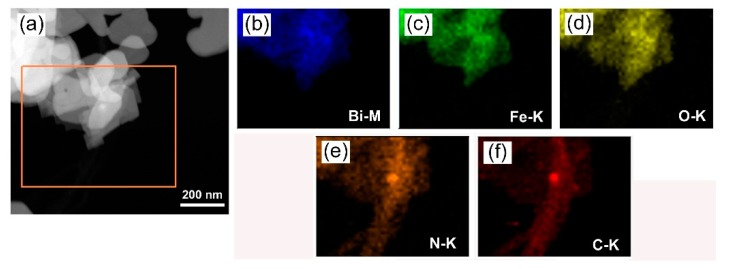
(**a**) DF-STEM image of 8%g-C_3_N_4_/1.5%CNT/BFO; and (**b**–**f**) the energy dispersive X-ray elemental mapping images of the selected region in (**a**).

**Figure 7 micromachines-09-00613-f007:**
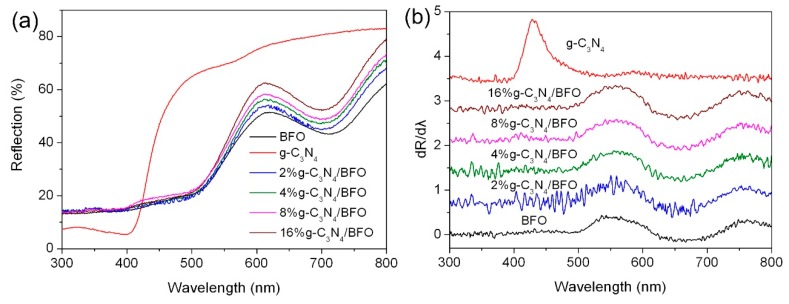
(**a**) UV-vis diffuse reflectance spectra of BFO, g-C_3_N_4_ and g-C_3_N_4_/BFO composites; and (**b**) the corresponding first derivative curves of the diffuse reflectance spectra.

**Figure 8 micromachines-09-00613-f008:**
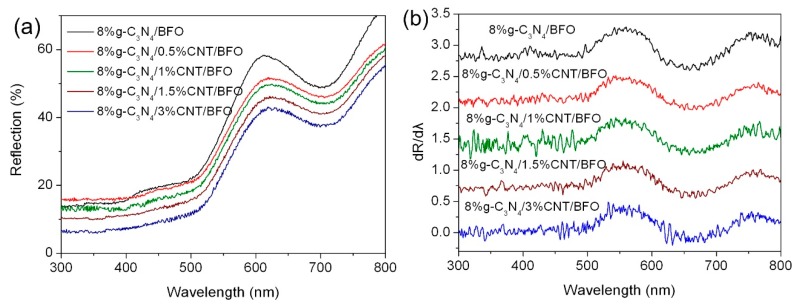
(**a**) UV-vis diffuse reflectance spectra of g-C_3_N_4_/CNT/BFO composites; and (**b**) the corresponding first derivative curves of the diffuse reflectance spectra.

**Figure 9 micromachines-09-00613-f009:**
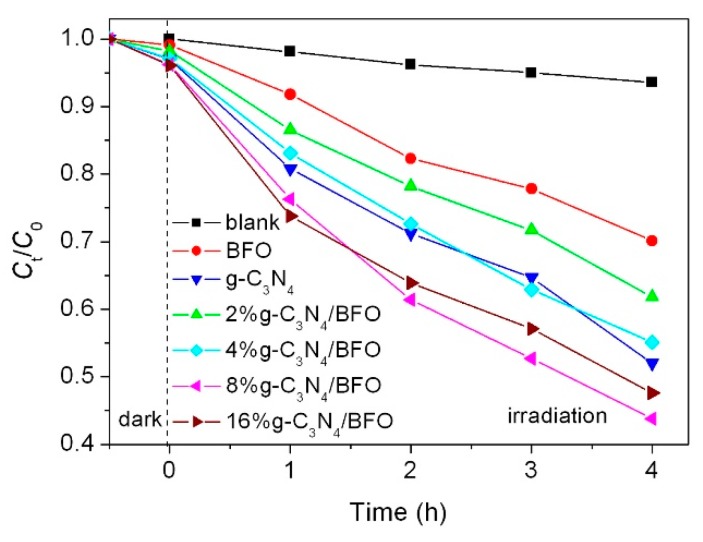
Photocatalytic activities of g-C_3_N_4_, BFO and g-C_3_N_4_/BFO composites toward the degradation of AO7 under simulated sunlight irradiation, along with the blank and adsorption experiment results.

**Figure 10 micromachines-09-00613-f010:**
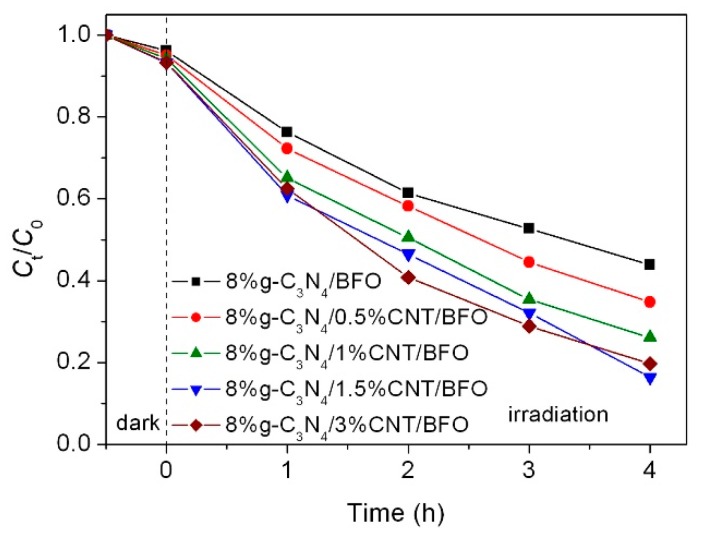
Photocatalytic activities of g-C_3_N_4_/CNT/BFO composites toward the degradation of AO7 under simulated sunlight irradiation, along with the adsorption experiment result.

**Figure 11 micromachines-09-00613-f011:**
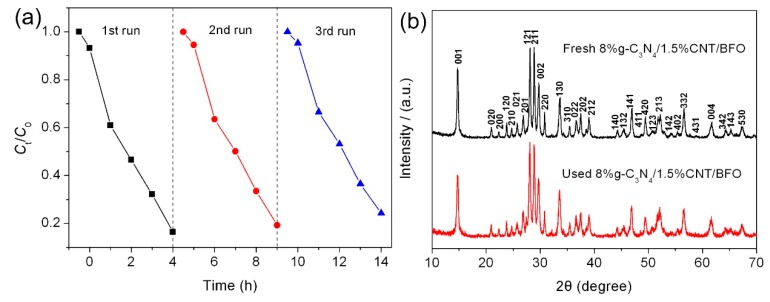
(**a**) Photocatalytic recyclability of 8%g-C_3_N_4_/1.5%CNT/BFO for the degradation of AO7 under simulated sunlight irradiation; and (**b**) XRD pattern of the recycled 8%g-C_3_N_4_/1.5%CNT/BFO.

**Figure 12 micromachines-09-00613-f012:**
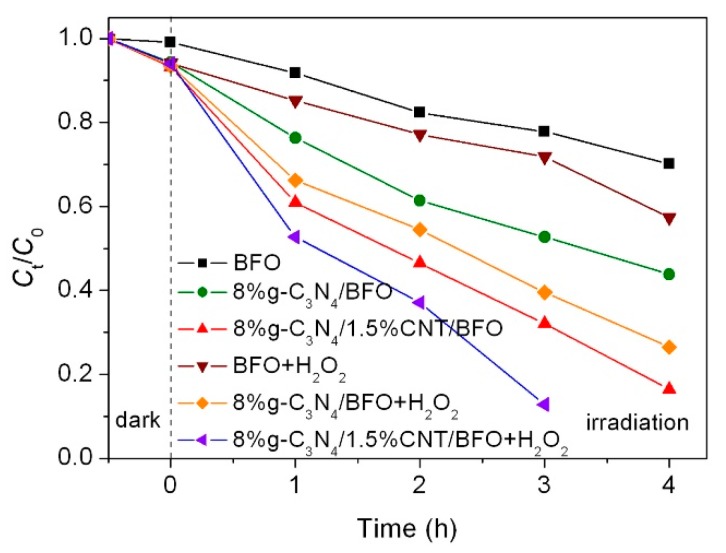
Photo-Fenton-like catalysis properties of BFO, 8%g-C_3_N_4_/BFO and 8%g-C_3_N_4_/1.5%CNT/BFO under simulated sunlight irradiation and in the presence of H_2_O_2_.

**Figure 13 micromachines-09-00613-f013:**
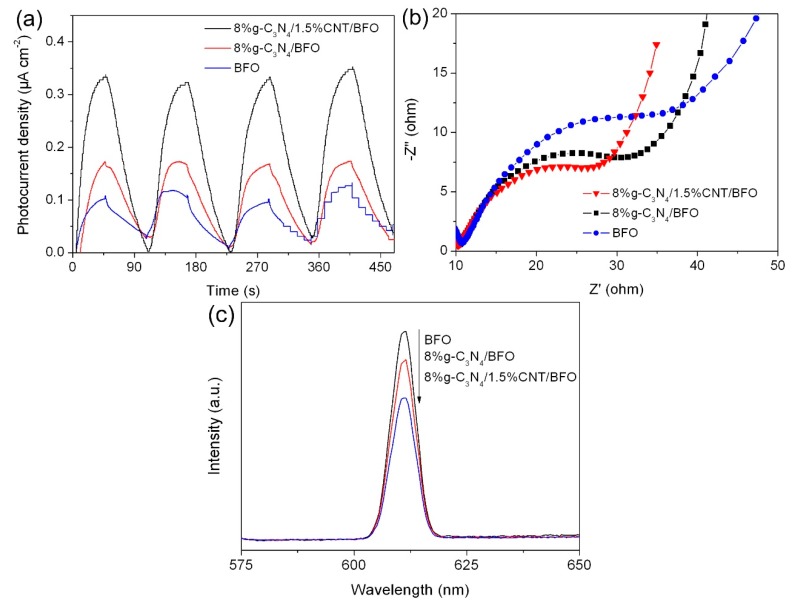
Photocurrent responses curves (**a**); EIS spectra (**b**); and PL spectra (**c**) of BFO, 8%g-C_3_N_4_/BFO and 8%g-C_3_N_4_/1.5%CNT/BFO.

**Figure 14 micromachines-09-00613-f014:**
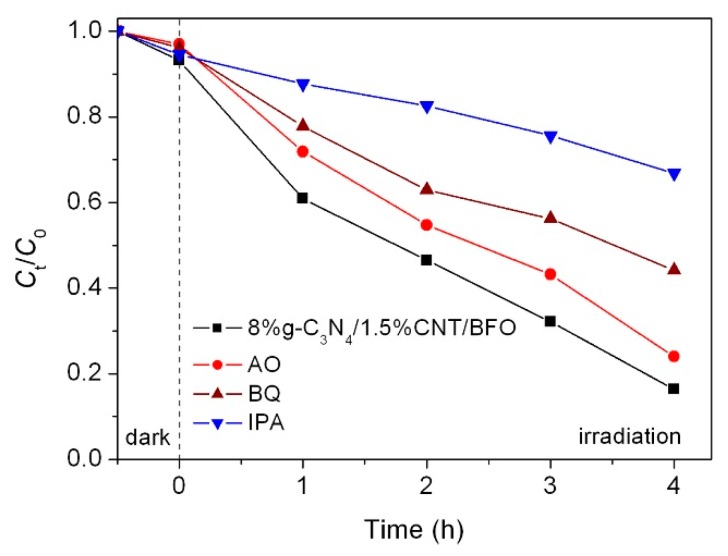
Effects of IPA, BQ and AO on the photocatalytic degradation of AO7 over 8%g-C_3_N_4_/1.5%CNT/BFO.

**Figure 15 micromachines-09-00613-f015:**
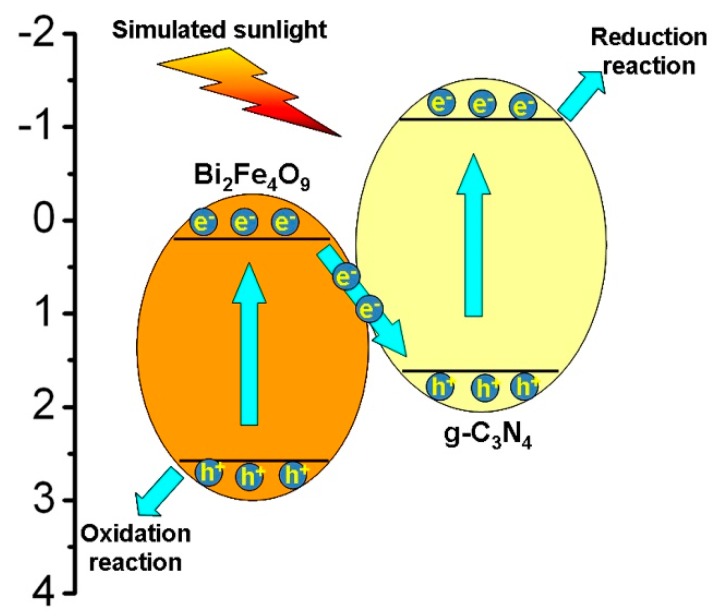
The proposed photocatalytic mechanism of g-C_3_N_4_/BFO composites.

**Figure 16 micromachines-09-00613-f016:**
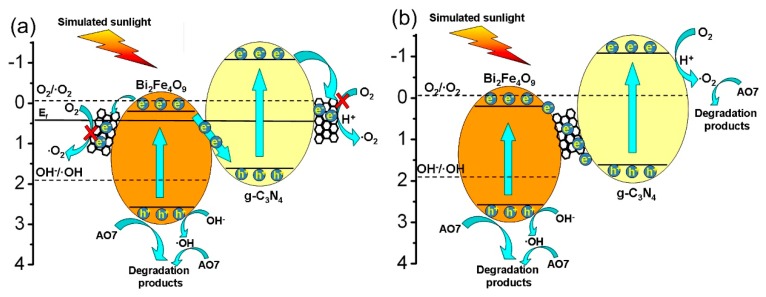
Schematic illustration of two possible mechanisms for charge migration and photocatalysis in g-C_3_N_4_/CNT/BFO composites. (**a**) CNT on the surface of g-C_3_N_4_ or BFO; (**b**) CNT between g-C_3_N_4_ and BFO.

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
