# Peer review of "Construction of Z-Scheme g-C3N4/CNT/Bi2Fe4O9 Composites with Improved Simulated-Sunlight Photocatalytic Activity for the Dye Degradation"

_micromachines, 2018, doi:10.3390/mi9120613_

Reviewer 1 Report

The manuscript entitled “Construction of Z-scheme g-C3N4/CNT/Bi2Fe4O9  composites with improved simulated-sunlight  photocatalytic activity for the dye degradation” presents synthesis of ternary all-solid-state Z-scheme g-C3N4/CNT/Bi2Fe4O9 composites with enhanced photocatalytic activity by a  hydrothermal method and using of obtained composites for degradation of acid orange 7 under simulated sunlight irradiation. The authors characterized obtained composites with many methods: FTIR, XRD, SEM, TEM, BET and particle size distribution analysis. They demonstrated that the g-C3N4/CNT/BFO composites exhibit remarkable enhanced photocatalytic activity compared with bare BFO and g-C3N4/BFO composites and proposed the photocatalytic mechanism of  studied composites according to the photo electrochemical measurement, photoluminescence, active species trapping experiment and energy-band potential analysis.

The obtained results are interesting, interpretation of results and references are correct,

I  recommend this manuscript to publication with minor revision.

My comments are following:

3.1.XRD and  FTIR analysis

in Fig. 2 should be selected bands, which are described in the text, because it is difficult to determine their exact location based on the scale

line 173, 175-177 in the same chapter – should be “ absorption peak”, not “adsorption peak”

the interpretation of TEM image of 8%g-C3N4/1.5%CNT/BFO presented in text is, in my opinion, not completely confirmed by the image.

Author Response

(1) In Fig. 2 should be selected bands, which are described in the text, because it is difficult to determine their exact location based on the scale.

Responses: Thank the reviewer for giving this good suggestion. According to the reviewer’s suggestion, the “peaks” were corrected into “bands” and furthermore the absorption bands in the Fig. 2 were labeled.

(2) line 173, 175-177 in the same chapter – should be “absorption peak”, not “adsorption peak”

Responses: We are sorry for our negligence. According to the reviewer’s suggestion, we corrected “adsorption peak” into “absorption band”.

(3) the interpretation of TEM image of 8%g-C3N4/1.5%CNT/BFO presented in text is, in my opinion, not completely confirmed by the image.

Responses: Thanks. According to the reviewer’s comment, we changed the description “it can be seen that g-C3N4 nanoparticles and BFO nanoplates are well anchored with CNT, indicating that ternary heterojunctions are formed in the g-C3N4/CNT/BFO composites.” into “it can be seen that g-C3N4 nanoparticles and BFO nanoplates are connected with CNT, indicating the formation of ternary g-C3N4/CNT/BFO composites.”.

Reviewer 2 Report

I only have some minor and maybe picking question on the characterization.

Iron XPS is hard because of mulitipet splitting and shack-off peaks.  It maybe useful to reference a paper on FexOy and XPS. i.e. https://doi.org/10.1016/j.apsusc.2007.09.063 

The pore size analysis is BJH?  I know it is not BET!  There are different methods and the method should be referenced.  The tail out to extremely large pore doesn't seem physical, one need the method to judge.

Author Response

(1) Iron XPS is hard because of mulitipet splitting and shack-off peaks.  It may be useful to reference a paper on FexOy and XPS. i.e. https://doi.org/10.1016/j.apsusc.2007.09.063  

Responses: Thank the reviewer for giving this good suggestion. According to the reference provided by the reviewer, we have changed the description “The satellite peak at 718.2 eV is mainly ascribed to the multiple oxidation state of Fe” into “The peak at 718.2 eV is characterized as the corresponding satellite of Fe 2p3/2”. The main peak at 723.9 eV corresponds to the Fe 2p1/2 binding energy instead of Fe 2p3/2 binding energy. The good reference has been cited.

(2) The pore size analysis is BJH?  I know it is not BET!  There are different methods and the method should be referenced.  The tail out to extremely large pore doesn't seem physical, one need the method to judge.

Responses: Thank the reviewer for giving this good suggestion. Yes, the pore size analysis is based on the BJH method and the corresponding reference has been cited (Barrett, E.P.; Joyner, L.G.; Halenda, P.H. The Determination of Pore Volume and Area Distributions in Porous Substance. I: Computations from Nitrogen Isotherms. J. Am. Chem. Soc. 1951, 73, 373-380). I agree with the reviewer’s comment, the tail out to extremely large pore doesn't seem physical. Thus, we made the correction as follows.

The inset of Fig. 3 shows the pore-size distribution curves derived from the adsorption branch of the isotherm using the Barrett–Joyner–Halenda (BJH) method [53], implying the possible existence of micropores of 60–160 nm in the composite. The tail out to extremely large pore could not be caused by physical property.

Reviewer 3 Report

 The manuscript reported by Ribeiro et al. is an investiagation on the synthesis of g-C3N4/CNT/Bi2Fe4O9 3 composites with improved simulated-sunlight 4 photocatalytic activity for the dye degradation, based on the construction of Z-scheme 

It is a well-done and presented work, and deserves to be published in the present form

Author Response

The manuscript reported by Ribeiro et al. is an investiagation on the synthesis of g-C3N4/CNT/Bi2Fe4O9 3 composites with improved simulated-sunlight 4 photocatalytic activity for the dye degradation, based on the construction of Z-scheme .

It is a well-done and presented work, and deserves to be published in the present form.

Responses: We thank the reviewer for giving a completely positive comment on our manuscript.

Round  2

Reviewer 1 Report

I accept this manuscript in this present form

Reviewer 2 Report

This is a little awkward: The tail out to extremely large pore could not be caused by physical property.

Maybe: The tail out to extremely large pore is an artifact of the analysis?

Otherwize good.